# Let's just ask them. Perspectives on urban dwelling and air quality: A cross-sectional survey of 3,222 children, young people and parents

**Rachel Juel**[1], **Sarah Sharpe**[1], **Roberto Picetti**[1], **James Milner**[2], **Ana Bonell**[3,4], **Shunmay Yeung**[3], **Paul Wilkinson**[2], **Alan D. Dangour**[1], **Robert C. Hughes**[1]*

1 Department of Population Health, Centre on Climate Change and Planetary Health, London School of Hygiene & Tropical Medicine (LSHTM), London, United Kingdom, 2 Department of Public Health, Environments and Society, London School of Hygiene & Tropical Medicine (LSHTM), London, United Kingdom, 3 Department of Clinical Research, London School of Hygiene & Tropical Medicine (LSHTM), London, United Kingdom, 4 Medical Research Council Unit The Gambia at London School of Hygiene and Tropical Medicine, Banjul, The Gambia

* robert.hughes@lshtm.ac.uk

**Data Availability Statement:** All data collection tools, anonymised raw data, and data analysis code are available on the LSHTM Data Compass Site

## Abstract

This research aimed to capture and synthesise the views of children, young people, parents and expectant parents (CYPP) about the cities where they live, with a specific focus on air pollution (AP), in order to support the generation of evidence-informed policy that reflects CYPP's perspectives, ultimately contributing to the development of child-centered, healthier, sustainable cities. The Children, Cities and Climate (CCC) project used targeted social media adverts to recruit CYPP to complete an online survey with a combination of open and closed questions in order to collect perceptions about air quality in their home cities, the main sources of AP, and how they would improve their cities. The survey was completed by 3,222 CYPP in 59 of the most polluted cities in 14 countries. Nearly two in five (39%) CYPP cited AP as one of the worst things about their city, with motor transport perceived as the main contributor. CYPP reported differing views on whether their cities were becoming better (43%) or worse (34%) places to live (33% reported it was 'staying the same'). Numerous specific ideas to improve cities and urban air quality emerged, alongside an emphasis on also addressing structural barriers to change. A clear set of principles that should guide how city leaders act was also described, including the need to engage with young people meaningfully. CYPPs articulated good and bad experiences of urban living and perceived AP and traffic as pressing concerns. They provided a clear set of suggestions for improving their cities. Further efforts to engage young people on these issues are warranted.

## Introduction

The 2020 State of Global Air report found 100% of the world's population lived in areas that exceeded the recently tightened 2021 WHO air quality guidelines for fine particulate matter pollution ($PM_{2.5}$) [1]. These levels of air pollution (AP) pose serious health risks to children,

mirrored on Figshare. The dataset has been assigned a unique DataCite DOI. https://datacompass.lshtm.ac.uk/id/eprint/2654/ and Figshare DOI: 10.6084/m9.figshare.21947432.

**Funding:** This work was funded by the Fondation Botnar [OOG-21-006 to RH and AD]. This grant supported the salary (in-part) of all authors of this manuscript (RJ, SS, RP, JM, AB, SY, PW, AD, RH). The funders had no role in study design, data collection and analysis, decision to publish, or preparation of the manuscript.

**Competing interests:** RCH has received grants and/or been a paid adviser to charities The Children's Investment Fund Foundation and the Clean Air Fund for work related to health effects of air pollution and climate change.

and may reduce the life expectancy of a child born in 2019 by up to 12 months (compared to children born in the absence of $PM_{2.5}$ exposure) [2]. In 2016, the reduction in life expectancy translated to 6.5 million premature deaths due to household and ambient air pollution [3]. Those who are not killed by air pollution face increased risk of morbidity even before their birth, with maternal PM2.5 exposure linked to ~18% of global preterm births in 2017 [4]. Due to air pollution, these additional risks continue throughout early child development, adolescence, and young adulthood. Air pollution exposure has been associated with lung development and function in children, poor respiratory health in adolescence, and increased risk of lung cancer throughout the life course [4].

Though young people in both urban and non-urban places are impacted by air pollution, young people in urban areas are exposed to significantly higher levels than those in rural areas. In addition, in many settings (particularly low- and middle-income countries), as urbanisation accelerates, air pollution increases" [4]. Children currently represent 30% of the four billion people living in cities, but this will rise to 70% of 6.7 billion by 2050 [5]. Additionally, many of the sources of AP contribute to climate change, the effects of which children today will experience throughout their lives [6].

While the environmental and health impacts associated with AP exposure are increasingly well understood [7], little is known about how children, young people, parents and expectant parents (CYPP) perceive their urban environment and the quality of air in their cities. Additionally, the ideas and expectations of CYPP for improving urban air quality are under-explored.

Within the last five years, several reports have been published featuring youth voices, which sought to highlight the opinions young people have on their urban environments [8–11]. In the My City Too [8], young Londoners shared their aspirations for the city based on their lived experience. Though this visionary piece provided compelling ideas, no systematic attempt was made to understand young people's perception or experience of air pollution.

The Young Londoners' Priorities for a Sustainable City (9) mapped key issues reported by the young people against the sustainable development goals (SDGs). Within goal 11 of the SDGs (Sustainable Cities and Communities), environmental protection emerged as a key theme. Though air quality ranked second as the most important environmental issue, no recommendations for addressing air pollution were sought from the young people.

The Urban Work Index (10) utilized more systematic methods to survey and rank cities according to how young people rated different areas of urban living. Though this index highlights the beneficial aspects of each city in its current state, the young people were not asked to consider the impact of air pollution on their ratings.

The CYCLES report (11) provides profiles of seven majors cities throughout the world that describe the sustainability challenges faced by their young residents. These challenges were reported in several highly polluting sectors, including food, transport, home and energy use. Three of the seven cities included pollution in their description of one of these three polluting sectors. This narrative strengthens the evidence on the impact air pollution has on urban living, though again, no recommendations were made by the young people.

This research aimed to capture and synthesise the views of CYPP about the cities where they live, with a specific focus on AP, in order to support the generation of evidence-informed policy that reflects CYPP's perspectives, ultimately contributing to the development of child-centered, healthier, sustainable cities.

## Methods

This cross-sectional study collected data through an online survey. The survey was available to CYPP from any city in the world, though targeted recruitment (through paid social media

advertisements–S1 Appendix) was directed to CYPP in 16 major cities (Bhubaneswar, Dar es Salaam, Dhaka, Free Town, Glasgow, Harare, Jaipur, Lahore, London, Los Angeles, Mexico City, Milan, Nairobi, Quezon City, Quito, Tamale). These cities were chosen to reflect a variety of global settings, population sizes, and levels of development in both the Global North and South, and to include the hosts of the 2021 Conference of Parties (COP) 26 Summit (Glasgow and Milan). Eligibility for inclusion in the survey required respondents to currently live in a town or city, be a young person aged between 13 and 25 years old, or be a parent or expectant parent to a child aged <13 years old. A child was differentiated from a young person at the age of 13 due to internet regulations in most countries which require children under the age of 13 to seek parental approval for most internet use. Young people included those up to the age of 25. Though this is the standard age in many countries, it is not a globally recognised number. As this was a global survey, the cut-off point was selected by the research team in conjunction with youth advisers for the project. Detailed methods for population recruitment, pretesting methods, and data handling can be found in S2 Appendix.

## Data collection

The survey instrument (S3 Appendix) was designed and hosted on Typeform [12], an online survey platform. Participation in this survey was voluntary and respondents were informed they would not receive a reward for completion (for full statements, see S4 Appendix). Respondents self-selected their preferred language (from 10 options) and completed three demographics questions to check eligibility and city location. The survey questions included: 5 'closed' multiple choice questions to identify the 'best' and 'worst' parts of their cities (with the option to submit their own answer via an 'other' choice); to rank their city's air quality on a Likert scale [13] from zero ('awful & toxic') to ten ('amazing & fresh'); to suggest the main sources of AP in their city; and to indicate if they felt their city was becoming a better or worse place to live. Six 'open-ended' questions collected long-form comments about the CYPP's cities in general and the quality of the air. CYPP were asked to elaborate on previous answers and how they would improve their cities generally for young people and specifically regarding air quality.

## Informed consent

Informed consent was gained from all survey respondents. This was through completion of an online form as approved by the London School of Hygiene and Tropical Medicine (LSHTM) Ethics Committee (see S4 Appendix for consent questions). Given the data collected by the survey was exclusively non-identifiable and would be instantly anonymous, the Ethics Committee agreed this research low risk. In agreement with the Ethics Committee, we did not seek to inform or obtain consent from parents or guardians of child respondents (who were all 13 years old or older).

## Missing data

Only respondents with full demographic data who met the eligibility criteria, provided informed consent via an online form (approved by LSHTM Ethics committee—S4 Appendix), and successfully completed the survey were included in the full survey sample. An available case analysis method was used for each survey question, ensuring each question sub-population consisted of complete responses only. The demographics of subpopulation respondents were compared to the full sample's demographics to identify any patterns of item non-response.

Where possible, imputation methods were used to limit non-response bias for age and location using the following methods. Where age was not reported or reported as an unrealistic value in excess of 100 or less than 13 years old, age was recoded as 'missing' and survey respondents were analysed together. Where location was not reported or only reported by region or country, probable location was imputed based on the city featured in the recruitment advert clicked on.

## Data analysis

The data analysis plan (available on Data Compass [14]) was developed in advance of data collection. In short, descriptive statistics were completed for the full sample and subpopulations by city and for each survey question. The quantitative and qualitative survey responses were analysed for the full sample and results were stratified by age, whether the respondent was a young person or parent/expectant parent, and by the PM2.5 pollution in their city.

Where there were fewer than five responses from an individual city, these were clustered based on location linked to the advertisement they responded to. Countries were categorised as high or middle and low income as identified by the Organisation for Economic Co-operation and Development [15]. Parents (n = 509) and expectant parents (n = 283) were combined in analyses. Where relevant, 'other' responses to closed questions were coded against existing options, and when appropriate options didn't exist in the survey instrument, answers were analysed thematically with the other qualitative answers.

Analysis of the quantitative data was completed using write Stata Statistical Software: Release 17 (College Station, TX: StataCorp LLC). The demographics (age and target group) of the full sample and each subpopulation by question were described using averages and frequencies. The sample was stratified using these demographics prior analysis of the quantitative questions. Responses to multiple choice questions were summarised using frequencies while the Likert scale responses were reported using the mean and standard deviation.

Reponses to the qualitative questions were analysed together using an interpretive grounded theory approach. Free text responses were translated into English by RJ using Google Translate and were not edited for grammar or punctuation. Quotations that included identifiable information were redacted before analysis. These responses were coded (by RJ) in NVivo 12 [16] using initial open coding methods to identify and group important words or groups of words, with a focus on the underlying meaning of responses. Coding was regularly discussed and reviewed with RH, who also reviewed open-ended questions. Where possible, in-vivo codes were used to capture the participant's words as representative of a broader concept. Intermediate codes were created to identify core categories drawing on initial coding. Analysis was undertaken simultaneously with data coding to look for emerging concepts and groups of themes through a series of meetings between RJ and RH, and the wider team. RJ and RH are both public health and climate change researchers. RH's particular expertise is in AP and early childhood development, while RJ focuses on food systems and planetary health. The expertise of the wider team comprises pediatric medicine, epidemiology, mathematical modelling, maternal health, and research communications. Where data saturation became apparent and no new codes or groupings of codes were identifiable, coding was ceased. Finally, a conceptual framework was developed to synthesise the themes and sub-themes which emerged.

## Patient and public involvement

This project involved local collaborators in Kenya and Dar Es Salaam. Our partner in Kenya and Dar Es Salaam, Shujaaz Inc., contributed to survey method pretesting and translation into the Sheng local dialect of Swahili. They additionally contributed to recruitment to our online

survey, through neutral pushes to their viewer-base. Finally, they assisted in dissemination of research findings throughout East Africa and at a variety of events hosted by LSHTM. Additional collaborations involved the Zimbabwe LSHTM Research Partnership, and a group of young people throughout the world. These collaborations similarly occurred during the research dissemination, with partners contributing to several LSHTM panel-events both in-person in the United Kingdom at the COP26 and Conference of Youth (COY) 16 climate change conferences and online.

### Ethical approval

Ethical approval was granted by the LSHTM Research Ethics Committee (LSHTM Climate change, cities, child and young people's health: children and young people survey and engagement, reference 26313). In addition, municipal authorities in all 16 focal cities were informed about the planned survey and were invited to provide feedback and offered a briefing on the results.

### Results

In total 3,808 eligible and consenting participants were recruited to the study and 3,222 (85%) of these completed the online survey. On average, respondents clicked on the survey link after 4.3 views of a recruitment advertisement, but this ranged from an average of 1.3 views in London, United Kingdom (UK) to 17.1 views in Freetown, Sierra Leone. The survey was built to take less than 10 minutes to complete, however, Typeform allows respondents to leave and return to a survey an unlimited number of times using the same link prior to submission. Subsequently, the median survey completion time was 8.7 minutes (range: 0.5–458.8 minutes). Responses were not excluded from the analysis for outlying times.

The survey respondents reported being from 14 different countries and 59 cities (Fig 1, S5 and S6 Appendices). Most responses (in part due to how the advertisement targeting algorithm prioritised lower cost per click markets) were from highly polluted cities in low- and middle-income countries (S7 Appendix). The mean age of children and young people sample was 20 (Standard deviation (SD) 3.45), and amongst parents/expectant parents was 34 (SD 9.25); 75% of all respondents (n = 2,430) were young people aged 13–25 years old (mean age 21, SD 4.3);

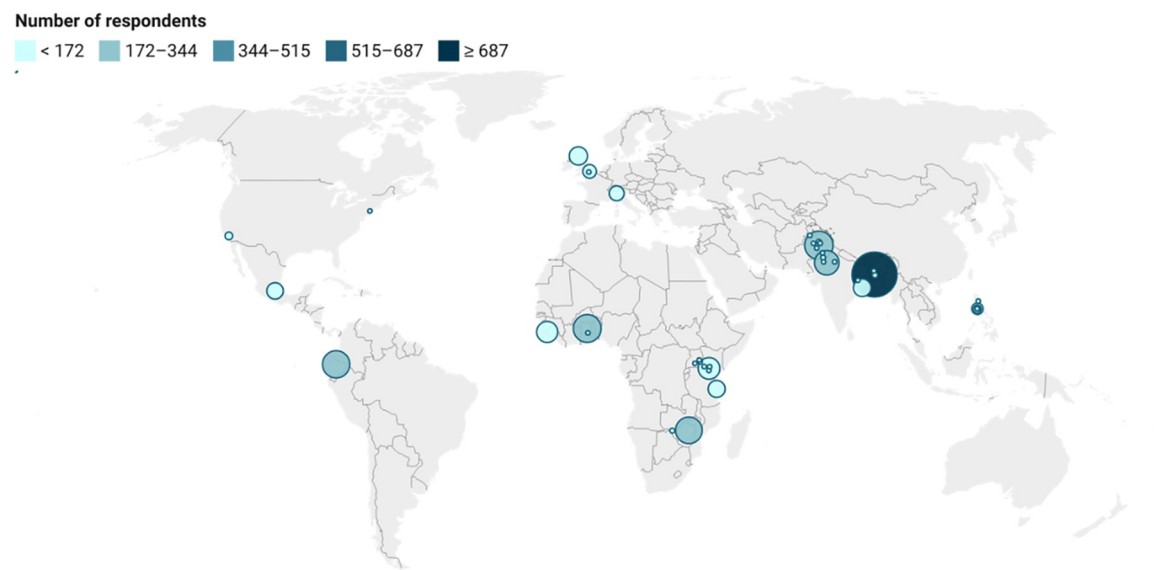

**Number of respondents**

< 172 | 172–344 | 344–515 | 515–687 | ≥ 687

**Fig 1. Distribution of survey responses by self-reported city (n = 59).**

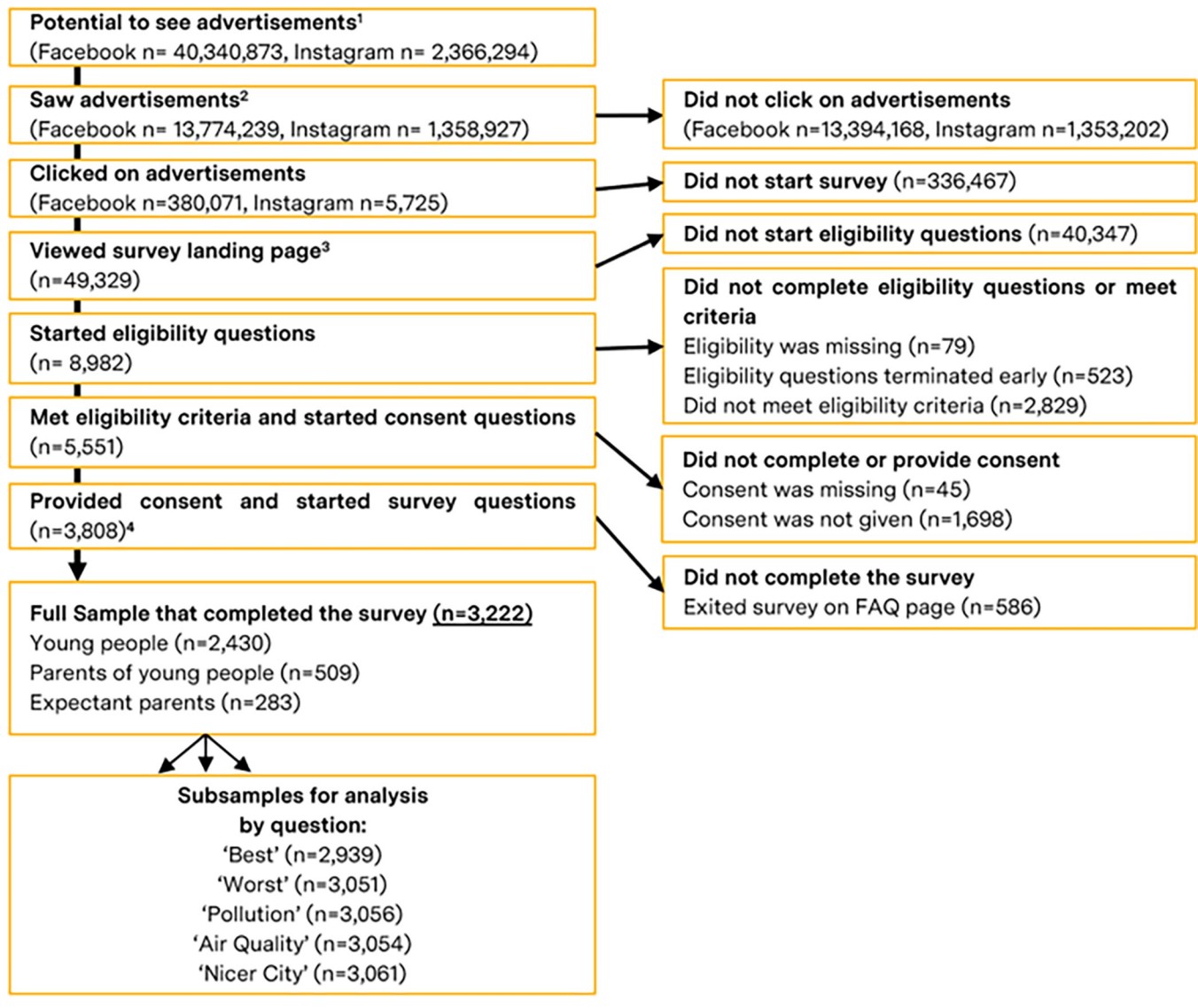

**Fig 2. Creation of full sample and subpopulations.** [1] The number of individual social media users on each platform that fit the desired demographics of the target population and could be *potentially* shown the advertisement during the paid advertising campaign. [2] The number of social media users on each platform that were shown an advertisement during the campaign. [3] The number of social media users that clicked on the advertisement on either social media site and were successfully transferred to the survey landing page on Typeform.com to begin eligibility and informed-consent questions. Not all social media users targeted by the paid advertising campaign were eligible or gave consent to participate in the survey. [4] The number of eligible and consenting survey participants who could *potentially* complete the full survey.

16% (n = 509) were parents of children aged <13 (mean age 39, SD 7.8); and 9% (n = 283) were expectant parents (mean age 32, SD 7.3); this did not differ for any question's subpopulation (S8 Appendix). The distribution of survey respondent's demographics by city can be found in S9 Appendix.

A flow-chart of sample formation can be found in Fig 2.

## Overall perceptions about their cities

Initially, respondents were asked about their cities in general, followed by specific questions on air quality. The most common "best" aspects reported were proximity to family (reported by 30%); the diversity of activities (27%); and proximity to school or work (27%); the most

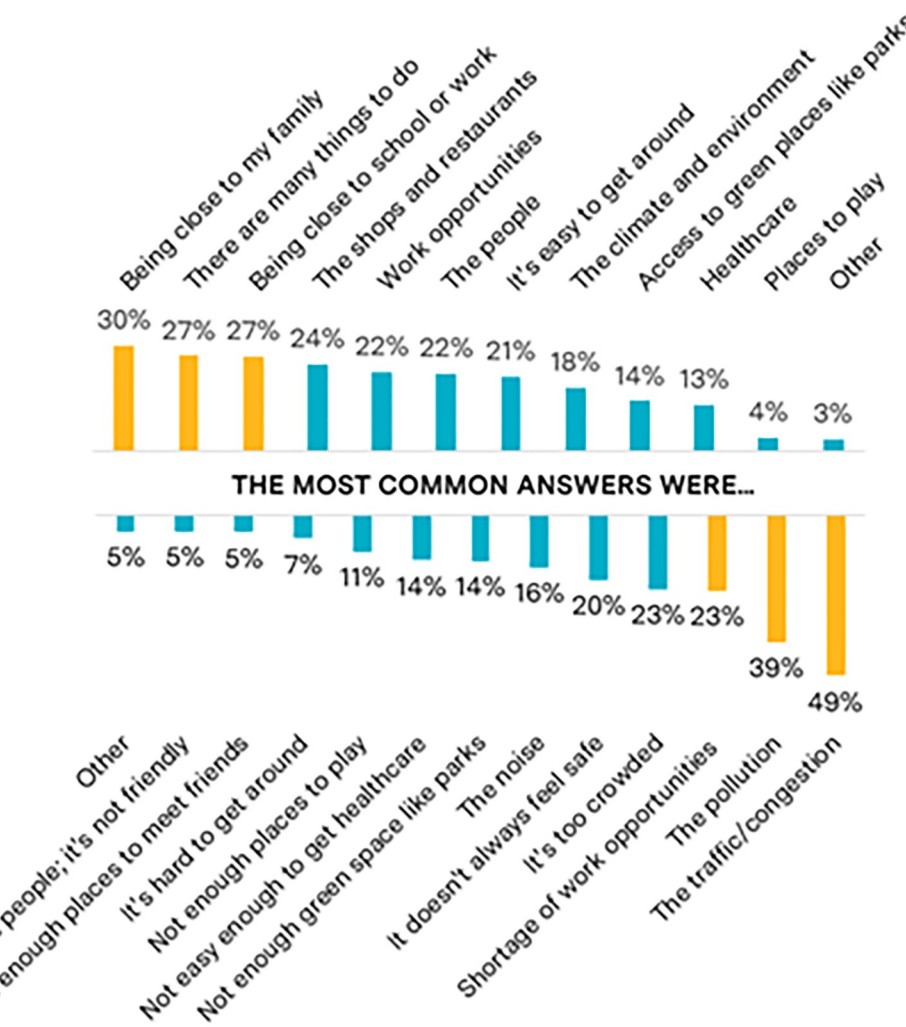

**Fig 3. The percentage of n = 2,939 and n = 3,051 respondents that reported each item within the top 3 'best' and 'worst' aspects of their cities.**

common "worst" things were traffic congestion (reported by 49%); pollution (39%); and a shortage of work opportunities (23%) (Fig 3). No trends were observed when the best and worst aspects of cities were stratified by age, PM$_{2.5}$ quartile, or respondent group (S10 and S11 Appendices). 43% of the sample reported that their cities were "improving", 23% said there was no change and 34% reported their cities were getting worse (S12 Appendix).

## Perceptions on air quality and major causes of AP

The mean air quality rating (on a scale of 0 ('awful & toxic') to 10 ('amazing & fresh') reported by 3,054 respondents was 5.1 (sd 2.9). There was no obvious association between self-rated air quality and quartile of air quality of cities; the mean score in the least polluted cities was 4.71 (SD 2.9) and in the most polluted was 6.01. No trends were identified when stratifying by age or respondent group (S13 Appendix).

The most commonly reported sources of AP were motor transport, factories, rubbish burning and construction/building work. Agriculture/farming, household cooking and household heating were all reported by smaller proportions of respondents (2%, 5% and 5% respectively); all considerably lower than AP blown in from outside the city (8%). 'Other' responses included natural sources such as dirt and dust, but were mostly based on human actions (burning of waste/sewage) and specific man-made objects (aeroplanes, plastic shopping bags). No clear patterns were identified when stratifying the major sources of AP by $PM_{2.5}$ quartile, major city, eligibility criteria, or age of respondent (S14 Appendix).

### Thematic analysis of open-ended questions

Three sets of themes emerged from the analysis of qualitative survey responses. An overall summary of the three themes and their respective subthemes is illustrated in Fig 4.

The first theme that emerged was a set of **specific ideas or asks** representing distinct changes or policy priorities to curb AP and improve cities more generally for young people. Sub-themes identified included 'city design and space', 'urban mobility', 'health', 'education', 'skills and jobs', and 'other basic services'. The breakdown of these ideas into sectors with exemplary quotes is provided in S15 Appendix.

A second emerging theme concerned **structural barriers that need to be addressed** before these specific ideas and asks can be achieved. These subthemes were felt to represent broader, more fundamental issues faced by cities and were 'inequality', 'corruption and bad governance', 'lack/absence of young people in decision making', and 'a lack of consciousness for the environment', listed with illustrative quotes in S16 Appendix.

The third set of themes described **Guiding principles for *how* to act** to address the structural barriers and to achieve the specific ideas and asks of urban young people. These principles were to 'be ambitious and creative', 'engage young people in meaningful ways', and to 'understand and tackle inequalities'. Illustrative quotes for each of these are included in Box 1.

---

### Box 1: Illustrative quotes of guiding principles for how to act, by subtheme

**Ambitious and creative**

- *"I would also have wifi hotspots citywide for children and young people who have no access to wifi at home so that they can complete work and not fall behind."*

- *"build arcades, amusement parks etc. Just want them to enjoy and prosper in life."*

- *"Plants on each rooftop"*

**Engage young people meaningfully**

- *"consider the opinions of the children and young people to facilitate development in our country"*

- *"I really hope the government will embrace the opinions from us, the children and young people..."*

**Understand and tackle inequality**

- *"The rich don't care about the poor, and the high ranking politicians don't care for the masses as well but only their families, which is 95% leading to corruption, wherein they don't give job to the capable graduated ones except their love ones and families from the same ruling party"*

---

- *"I'd hopefully like to make an equal society where there is mutual respect for everyone"*

- *"Positive change needs to happen now. Joined up thinking, to implement a masterplan that takes the whole city forward, rather than the occasional isolated success story"*

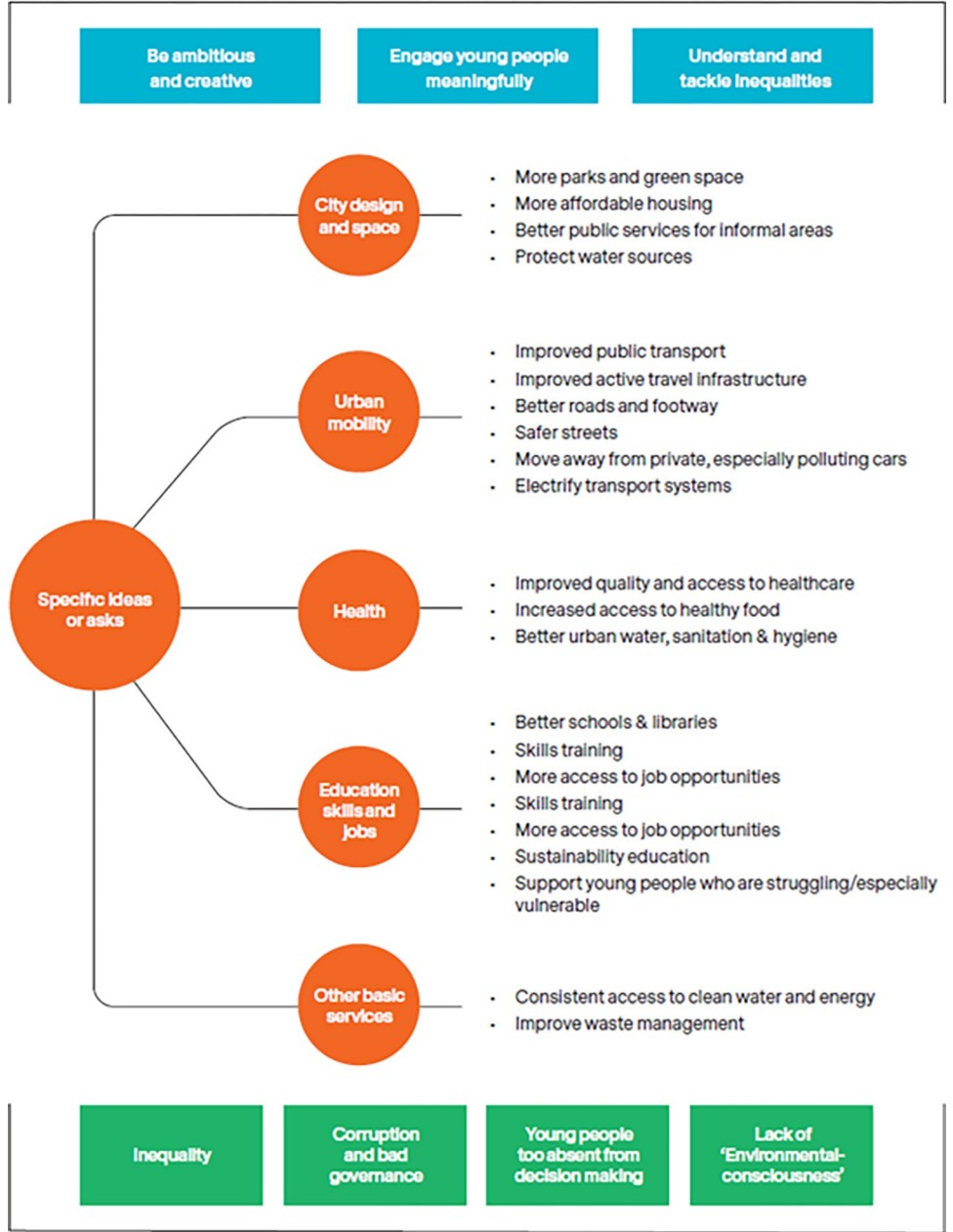

**Fig 4. Summary of the thematic analysis findings.**

## Discussion

In this survey, urban children, young people and parents in this survey provided important insights into the lived experience in their cities. Commonalities of the 'best' and 'worst' aspects of cities emerged, and were consistent across all respondent groups, stratified by age and $PM_{2.5}$ quartile. No clear majority was reached about the state of change in cities; respondents both felt their cities were improving and becoming worse. A common theme amongst those who responded to our survey was a high level of concern about AP, often accompanied by a broad set of ideas for how urban air quality could be improved. Additional recommendations were made to improve cities more generally for children and young people. To achieve these improvements, several notable barriers were identified that must be addressed, and a set of principles were developed to guide decision maker's actions.

### Strengths of this study

The use of social media recruitment and online surveying garnered a large number of responses, from a broad variety of cities. A combination of closed and open-ended questions allowed both quantitative and qualitative insights to be drawn, enabling us to explore where AP sits amongst a broader set of concerns and a rich set of underlying themes and ideas.

Recruitment through social media may have allowed for the inclusion of people who are not easily recruited through 'analogue' methods or are often systematically excluded from research panels (e.g. under 18, lower income, lower consumption groups). Additionally, the online survey instrument could be directly completed by young people, rather than by their parent/carer, or their school which limited misclassification of ideas which may occur when provided by a proxy adult.

The survey instrument was available for completion in 10 languages and was optimised using iterative pre-testing and translation checking by native speakers. The selected languages enabled global participation, as they are spoken as a first language by 40% of the global population, and are all (excluding Swahili) in the top 22 most commonly spoken languages worldwide [17].

### Limitations

Selection bias is likely to have occurred due to a combination of factors. Firstly, using online recruitment data collection systematically excludes those who do not have access to the internet, a computer or mobile device, and those who do not use social media platforms, meaning that our sample may have been biased towards more affluent people who can afford to be online and utilise such devices, and those who spend more time on social media.

Secondly, the costs per click for online recruitment advertisements varied by location, and those areas with lower cost per click were disproportionately advertised to due to the algorithms employed by the social media networks (despite attempts to mitigate this). The sample is therefore biased towards low- and middle-income countries, more polluted cities and especially those in South Asia (48% of the sample was from South Asia and over half the South Asian population from Dhaka, Bangladesh). Although the recruitment adverts were deliberately 'generic', aiming to appeal to all CYPP living in the city and the survey was short to reduce the risk of drop out, the topic of the survey is likely to have appealed more to certain groups of people (those who are interested in the climate/environment/urban living/child health).

Finally, though young people were involved in the development and pre-testing of the survey questions and discussion and dissemination of the results, they were not involved in the formal analysis or write up of the survey results.

## Putting this study into context

This survey was one of the largest, most geographically diverse of its kind, seeking to highlight the perspectives of global CYPP on urban living and AP. Several surveys were identified that more generally sought out the priorities of urban youth for improving how they live, work and play in cities [8–11], though none of these surveys explicitly sought CYPP perspectives on the air they breathe, or their recommendations for improving their city's air quality. Additionally, though several of these surveys were comparable in size [8–10] (n = 1,000, n = 2,002, and n = 9,000 respectively), only two included the voice of CYPP in low- and middle-income countries (LMICs) [10, 11], and neither provide a rich narrative of the challenges faced by CYPP in LMICs nor specific and discernable recommendations.

One survey of 1000 young people aged 12–19 in London identified some similar priorities for city improvements, including creating "car free streets" and "more beautiful and green places" [8] both of which emerged from our study also. In another survey of 2,002 young people aged 16–24 in London, 30% of respondents ranked AP in the top three most pressing issues to be addressed by their local authority [9]. AP was the fourth most prioritized issue out of all social, environmental, and economic issues. Within the environmental issues, improving air quality was the second-most prioritised issue, following only improvements in recycling and waste reduction. Although these young people prioritised environmental issues and independently demonstrated a level of environmental consciousness, they also noted the need for increased environmental education, corroborating a systemic lack of environmental consciousness remains a barrier that needs to be addressed in cities.

A global survey of 9,000 young people aged 15–29 highlighted some themes consistent with our survey, most notably the importance of promoting environmental awareness alongside delivery of specific policies [10]. Another study that included both LMICs and high-income countries (HICs), identified similar barriers young people aged 12–24 face to living sustainably to those reported in our survey. These included growing inequality and poor availability of public services and safe, efficient public transport [10].

## Implications of this research for policy makers

There is obvious need for acceleration of efforts to address urban air pollution: the importance which young people place on this issue is clear. This is not surprising in the context of school closures due to extreme pollution events in South Asia [18, 19]. These extreme events, in combination with the known health consequences of AP, led UNICEF to declare climate change and AP to be a child's rights crisis (3)

Secondly, the rich insights and ideas shared by participants in this study suggest that policy makers should make greater effort to engage CYPP in urban and air pollution policy discussions. Both Rights-based and pragmatic arguments for this can be convincingly made given the United Nations Convention of the Rights of the Child (which asserts their Right to be involved in developing solutions to the challenges they face [20] and the creative ideas they report in this study).

## Unanswered questions and future research

Future research into this issue could employ better methods to improve sample representativeness (e.g. combining online and offline surveying with particular efforts to include marginalised communities (e.g. homeless and migrant youth and those in informal settlements)) as air pollution levels and youth's awareness of air pollution will differ depending on the social infrastructure, and other determinants. Further research on the experience of urban youth is

needed across all socio-economic levels, races, and must include cities of all economic statuses and levels of urbanization.

Additionally, richer perspectives could be gained from collecting narrative via focus groups or urban co-design workshops with CYPP in cities. Co-development and leadership of these focus groups with CYPP would allow for deeper and more genuine insights to be collected regarding the lived experience of urban children.

Finally, in this time of rapidly changing climate, future research is needed to understand the perspectives of urban CYPP on other environmental health and climate-related issues, including, for example, exposure to urban heat, different models of urban mobility and shifting urban food systems. The health and non-health effects of these shifts must also be understood, given the impact they are likely to have on the health of the next generation.

## Conclusion

Young people (and parents) living in cities around the world clearly articulate both what is good, and what is bad, about growing up and living in their city. Subsequently, these 16 cities were not consistently reported as becoming 'better' or 'worse' places to live. Many beneficial qualities of urban living were noted, particularly proximity to family, though those AP and traffic were perceived as pressing concerns, particularly by those living in polluted cities. In addition, young people, when asked, provide a clear set of suggestions for how their cities can be improved. These recommendations were multi-sectoral, including city design, and multiple aspects of urban living. Though recommendations were specific, implementation of these ideas may be prevented by structural barriers including bad governance. Young people recommended a set of principles to guide decision makers to overcome these barriers and implement their specific ideas. Notably, young people requested to be consulted more in all matters that pertained to them. This survey attempted to involve young people in all stages of mechanism design and research dissemination, and demonstrated that online surveying can be an efficient, albeit imperfect, way to engage young people. Further, more in-depth, efforts to engage young people on these issues are warranted.

### Data storage and sharing

All responses were stored on the Typeform.com servers (hosted in a Virtual Private Cloud by Amazon Web Services) and processed and analysed on password-protected encrypted devices. All data collection tools, anonymised raw data, and data analysis code are available on the LSHTM Data Compass Site [14] mirrored on Figshare [21]. The dataset has been assigned a unique DataCite DOI [22]. Confidentiality of survey responses was maintained throughout; no personally identifiable data was collected. Age in years was collected, and the location was recorded at the city level. In addition, no IP address or other locator/identifiers (like cookies) were captured.

### Supporting information

**S1 Appendix. Most successful advertisement for each target city.**
(DOCX)

**S2 Appendix. Detailed methods for population recruitment, pretesting methods, and data handling population recruitment.**
(DOCX)

**S3 Appendix. Survey instrument.**
(DOCX)

**S4 Appendix. Language, consent, and eligibility questions.**
(DOCX)

**S5 Appendix. Survey completions by country.**
(DOCX)

**S6 Appendix. Respondents by self-reported town.**
(DOCX)

**S7 Appendix. Distribution of respondents (n = 3,222) and focal cities (n = 16) by pm2.5 and income indicators.**
(DOCX)

**S8 Appendix. Demographics for full sample and each subpopulation.**
(DOCX)

**S9 Appendix. Distribution of respondent demographics in total (n = 3,222) and by focal city.**
(DOCX)

**S10 Appendix. Percentage of total n respondents that reported each item within the top 3 'best' aspects of their cities, stratified by PM2.5 quartile, age bucket, and respondent group.**
(DOCX)

**S11 Appendix. The percentage of total n respondents that reported each item within the top 3 'worst' aspects of their cities, stratified by PM2.5 quartile, age bucket, and respondent group.**
(DOCX)

**S12 Appendix. The percentage of total n respondents that reported better, worse, or no changes to their cities, stratified by PM2.5 quartile, age bucket, and respondent group.**
(DOCX)

**S13 Appendix. Mean air quality (scale 0–10) reported by n = 3,054 respondents, stratified by PM2.5 quartile.**
(DOCX)

**S14 Appendix. The percentage of total n respondents who reported each major source of AP in their city, stratified by PM2.5 quartile, age bucket, and respondent group.**
(DOCX)

**S15 Appendix. Illustrative quotes of youth's specific ideas and asks for their cities, by subtheme.**
(DOCX)

**S16 Appendix. Illustrative quotes of structural barriers to change, by subtheme.**
(DOCX)

## Acknowledgments

Martha Jennings, Tori Griffiths, and Kirsten Dawes provided substantial support to the implementation of the project. Without them, this research could not have been completed, and we thank them for their valuable contributions. Thank you to the young people who completed the

survey and shared insights into their lives, these were invaluable. Thank you to those at Empower Agency, Shujaaz Inc., YOUNGO, and all other organisations that advised and informed the design and administration of the survey and contributed to public engagement and research dissemination. We would like to acknowledge and appreciate the support Meta has provided for this survey, in the form of advertisement credits used for recruiting participants to this survey.

## Author Contributions

**Conceptualization:** James Milner, Shunmay Yeung, Alan D. Dangour, Robert C. Hughes.

**Data curation:** Rachel Juel, Robert C. Hughes.

**Formal analysis:** Rachel Juel, Robert C. Hughes.

**Funding acquisition:** Sarah Sharpe, Alan D. Dangour, Robert C. Hughes.

**Investigation:** Rachel Juel, Roberto Picetti, Robert C. Hughes.

**Methodology:** Rachel Juel, Alan D. Dangour, Robert C. Hughes.

**Project administration:** Sarah Sharpe, Alan D. Dangour, Robert C. Hughes.

**Resources:** Sarah Sharpe.

**Supervision:** Robert C. Hughes.

**Visualization:** Robert C. Hughes.

**Writing – original draft:** Rachel Juel.

**Writing – review & editing:** Rachel Juel, Sarah Sharpe, Roberto Picetti, James Milner, Ana Bonell, Shunmay Yeung, Paul Wilkinson, Alan D. Dangour, Robert C. Hughes.

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
