## [Decision Letter · Decision Letter 0]

2 Jan 2023

PGPH-D-22-01277

Let’s just ask them.  Perspectives on urban dwelling and air quality: a cross-sectional survey of 3,222 children, young people and parents

Dear Dr. Hughes,

Thank you for submitting your manuscript to PLOS Global Public Health. After careful consideration, we feel that it has merit but does not fully meet PLOS Global Public Health’s publication criteria as it currently stands. Therefore, we invite you to submit a revised version of the manuscript that addresses the points raised during the review process.

We look forward to receiving your revised manuscript.

Kind regards,

Changwoo Han, M.D., Ph.D.

Academic Editor

Journal Requirements:

1. Our staff editors have determined that your manuscript is likely within the scope of our Climate Change and Human Health Call for Papers. This editorial initiative is headed by a team of Guest Editors for PLOS GPH: Renzo Guito (St. Luke's Medical Center College of Medicine, Philippines) and Tolu Oni (University of Cambridge) as well as the Guest Editor for PLOS Climate Anna Stewart Ibarra (Inter-American Institute for Global Change Research). The Collection will feature research that addresses all aspects of the intersection between climate and health, from the changing burden of communicable and non-communicable disease to the impacts of extreme events on health systems, as well as research that assesses potential adaptations to build healthier and more resilient societies. Additional information can be found on our announcement page: https://collections.plos.org/call-for-papers/climate-change-and-human-health/. 

If you would like your manuscript to be considered for this collection, please let us know in your cover letter and we will ensure that your paper is treated as if you were responding to this call.  Please note that being considered for the Collection does not require additional peer review beyond the journal’s standard process and will not delay the publication of your manuscript if it is accepted by PLOS GPH. If you would prefer to remove your manuscript from collection consideration, please specify this in the cover letter.

2. Please provide additional details regarding participant consent. In the Methods section, please ensure that you have specified (1) whether consent was informed and (2) what type you obtained (for instance, written or verbal). If your study included minors, state whether you obtained consent from parents or guardians. If the need for consent was waived by the ethics committee, please include this information.

3. Please send a completed 'Competing Interests' statement, including any COIs declared by your co-authors. If you have no competing interests to declare, please state "The authors have declared that no competing interests exist". Otherwise please declare all competing interests beginning with the statement "I have read the journal's policy and the authors of this manuscript have the following competing interests:"

4. Please amend your detailed Financial Disclosure statement. This is published with the article. It must therefore be completed in full sentences and contain the exact wording you wish to be published.

b. If any authors received a salary from any of your funders, please state which authors and which funders.

5. We have noticed that you have uploaded Supporting Information files, but you have not included a list of legends. Please add a full list of legends for your Supporting Information files after the references list.

Additional Editor Comments (if provided):

Please respond to the comments from the reviewers

Reviewers' comments:

Reviewer's Responses to Questions

**Comments to the Author**

1. Does this manuscript meet PLOS Global Public Health’s publication criteria? Is the manuscript technically sound, and do the data support the conclusions? The manuscript must describe methodologically and ethically rigorous research with conclusions that are appropriately drawn based on the data presented.

Reviewer #1: Yes

Reviewer #2: Yes

2. Has the statistical analysis been performed appropriately and rigorously?

Reviewer #1: Yes

Reviewer #2: No

3. Have the authors made all data underlying the findings in their manuscript fully available (please refer to the Data Availability Statement at the start of the manuscript PDF file)?

Reviewer #1: Yes

Reviewer #2: Yes

4. Is the manuscript presented in an intelligible fashion and written in standard English?

Reviewer #1: No

Reviewer #2: Yes

5. Review Comments to the Author

Reviewer #1: The contribution of the paper is general enough and informative for the audience of PLOS Global Public Health. However, some revisions have to be done.

1. The writing has to be revised for the readability of general readers. For example, abbreviations such as CYPP, and AP first appear without any explanations, though the original meanings are found subsequently in the text.

2. The figures are in poor resolution. It is hard to read especially Figure 1. Also, it would also be better if a table could be used to summarize the respondents' information in the 59 cities.

3. The main text reports the total number of young people, parents, and expectant parents respondents included in the survey data. However, it is better to report the distribution of numbers of different categories of respondents across the 59 cities. It may be that some of the cities have only a few parents included.

4. I wonder if some of the tables in the Supplementary Material can be moved to the main text. I believe they are more important to convey the findings rather than supporting material in the appendix.

5. The paper should put more effort in demonstrating the pioneering role of the survey. In lines 67-68 of the manuscript, it mentions that "the ideas and expectations of CYPP for improving urban air quality are under-explored". Then the authors cite a couple of papers without further detailed comparison with previous works.

Reviewer #2: Overview

Thank you for the opportunity to review the manuscript entitled “Let’s just ask them. Perspective on urban dwelling and air quality: a cross-sectional survey of 3,222 children, young people and parents". This interesting study has the advantages of capturing perceptions and views on air pollution in urban-dwelling CYPPs. However, a number of minor to moderate revisions, including revised interpretation of some findings, must be addressed.

I have the following comments and suggestions aimed to improve the quality of the paper

Please see below:

Figures

It is difficult to see the attached Figure files due to its low resolution. Please change the attached Figure files in high resolution.

Introduction

1. Line 58–60: The authors need to clearly describe which country and city it is.

2. I suggest that additional specific effect sizes of health effects such as disease burden and excessive mortality (or morbidity) caused by air pollution.

3. I propose adding a rationale for the importance of urban air pollution compared with non-urban areas (e.g., rural areas).

Methods

1. Line 86–88: What is the basis for the age definition corresponding to the inclusion criteria? I suggest that the authors will add references or sentences.

Data collection

1. Line 91–92: I wonder if the study participants were voluntary participation or there was no benefit.

2. I suggest that an indicator has been added to present the reliability and validity of the constructed questionnaire (e.g., Cronbach’s alpha).

Missing data

1. Line 119–120: The authors need to specify which imputation method was used. For the robustness of the research results, I suggest to compare the results before and after using the imputation method.

Data analysis

The authors need to clearly explain the data analysis.

Results

1. Line 185–187: On a typical person basis, how many minutes does the questionnaire take? If the response is too fast, or too slow, you may get distorted results. I propose to present the distribution of survey response time. In addition, I am curious about the analysis results after excluding both extreme values of the survey response time.

2. Line 192–192: I suggest that the average age is presented by separating children and young people.

Discussion

1. Line 363-385: I suggest that the authors compare previous studies by survey year, country or continent, race.

2. Air pollution levels and economic levels will be different in Europe, the United States, Asia, Africa and etc. The level of awareness of air pollution will vary depending on the social infrastructure and urbanization, including race, socioeconomic level, developing and developed countries. In the future, I hope that research will be conducted in more countries and global study will be conducted, and I suggest that related sentences be mentioned in Discussion.

6. PLOS authors have the option to publish the peer review history of their article (what does this mean?). If published, this will include your full peer review and any attached files.

**Do you want your identity to be public for this peer review?** For information about this choice, including consent withdrawal, please see our Privacy Policy.

Reviewer #1: No

Reviewer #2: No

---

## [Decision Letter · Decision Letter 1]

28 Feb 2023

Let’s just ask them.  Perspectives on urban dwelling and air quality: a cross-sectional survey of 3,222 children, young people and parents

PGPH-D-22-01277R1

Dear Dr. Hughes,

We are pleased to inform you that your manuscript 'Let’s just ask them.  Perspectives on urban dwelling and air quality: a cross-sectional survey of 3,222 children, young people and parents' has been provisionally accepted for publication in PLOS Global Public Health.

Best regards,

Changwoo Han, M.D., Ph.D.

Academic Editor

Reviewer Comments (if any, and for reference):

Reviewer's Responses to Questions

**Comments to the Author**

1. If the authors have adequately addressed your comments raised in a previous round of review and you feel that this manuscript is now acceptable for publication, you may indicate that here to bypass the “Comments to the Author” section, enter your conflict of interest statement in the “Confidential to Editor” section, and submit your "Accept" recommendation.

Reviewer #1: All comments have been addressed

Reviewer #2: All comments have been addressed

2. Does this manuscript meet PLOS Global Public Health’s publication criteria? Is the manuscript technically sound, and do the data support the conclusions? The manuscript must describe methodologically and ethically rigorous research with conclusions that are appropriately drawn based on the data presented.

Reviewer #1: Yes

Reviewer #2: Yes

3. Has the statistical analysis been performed appropriately and rigorously?

Reviewer #1: Yes

Reviewer #2: Yes

4. Have the authors made all data underlying the findings in their manuscript fully available (please refer to the Data Availability Statement at the start of the manuscript PDF file)?

Reviewer #1: Yes

Reviewer #2: Yes

5. Is the manuscript presented in an intelligible fashion and written in standard English?

Reviewer #1: Yes

Reviewer #2: Yes

6. Review Comments to the Author

Reviewer #1: I have no further comments.

Reviewer #2: (No Response)

7. PLOS authors have the option to publish the peer review history of their article (what does this mean?). If published, this will include your full peer review and any attached files.

**Do you want your identity to be public for this peer review?** For information about this choice, including consent withdrawal, please see our Privacy Policy.

Reviewer #1: No

Reviewer #2: No
